# Identifying Ecological Corridors of the Bush Cricket *Saga pedo* in Fragmented Landscapes

**DOI:** 10.3390/insects16030279

**Published:** 2025-03-06

**Authors:** Francesca Della Rocca, Emanuele Repetto, Livia De Caria, Pietro Milanesi

**Affiliations:** 1Department of Biology and Biotechnology “L. Spallanzani”, University of Pavia, Via Ferrata, 9, 27100 Pavia, Italy; francesca.dellarocca@unipv.it (F.D.R.); livia.decaria01@universitadipavia.it (L.D.C.); 2Department of Pharmacy and Biotechnology (FaBiT), University of Bologna, Via Belmeloro, 6, 40126 Bologna, Italy; 3Department of Biosciences, University of Milan, Via Celoria, 26, 20133 Milan, Italy; 4Department of Biological, Geological and Environmental Sciences, University of Bologna, Via Selmi, 3, 40126 Bologna, Italy; pietro.milanesi@unibo.it

**Keywords:** citizen science, insects, landscape connectivity, Omniscape, Orthoptera, site-occupancy models

## Abstract

The bush cricket *Saga pedo*, listed as Vulnerable by the IUCN, is severely threatened by habitat loss and fragmentation in Italy’s semi-natural grasslands. In our study, we used species distribution models and connectivity analysis to identify suitable habitats and ecological corridors in the northern Apennines. Our results showed that *S. pedo* prefers xerothermic grasslands with moderate woody vegetation and avoids areas characterized by intensive agriculture. We found that only 2.69% of the study area was suitable, highlighting the critical extent of habitat fragmentation. Our connectivity analysis showed that sustainable land management practices, including traditional agropastoral activities, maintaining grasslands with low woody vegetation, and minimizing intensive cropland expansion, are crucial for the survival of the species. In conclusion, we strongly advocate for immediate conservation measures such as restoring degraded habitats, enhancing ecological corridors, and promoting sustainable agricultural practices to ensure the conservation of this vulnerable species and its ecological role.

## 1. Introduction

Semi-natural grassland ecosystems play a crucial role in preserving biodiversity. The abundant variety of flora and fauna in these habitats largely relies on traditional farming and pastoral activities, which restrict plant competition, encourage the coexistence of diverse plant species, and support the survival of numerous uncommon and specialized insect species [1,2]. Nonetheless, semi-natural grasslands in Europe are experiencing a significant decline and fragmentation [3]. This is largely due to their conversion into high-yield crops or the cessation of farming and pastoral activities, such as mowing and extensive grazing. These changes have, in turn, favored the expansion of forest [4].

Among the inhabitants of semi-natural grasslands, the predatory bush cricket, *Saga pedo* (Pallas, 1771), is considered a high-priority species for conservation efforts. The species’ pronounced thermophily and heliophily lead it to inhabit xerothermic patches [5,6,7], making it a primary insect indicator of semi-natural grasslands [8]. Habitat fragmentation, by reducing the size and connectivity of these grassland patches, can lead to habitat degradation and a potential loss of suitable habitat [9,10]. This process is particularly critical for *Saga pedo* due to its limited mobility, which restricts its ability to traverse the increasing distances between isolated patches [11,12]. Therefore, preserving sufficiently large and connected grassland patches is essential for its long-term conservation.

Indeed, in the past it likely had a more uniform distribution [13]. However, nowadays, due to the decrease in the quality and availability of its suitable areas as well as the reduction in ecological connectivity between the remaining open patches, *S. pedo* is in sharp decline with localized and isolated populations [14]. For this reason, it is currently listed as Vulnerable globally by the International Union for Conservation of Nature (IUCN) in its 1996 assessment [15], while it is categorized as Least Concern at the European level in its 2016 assessment [16]. Additionally, it is included in Annex IV of the European Union Habitats Directive (1992), highlighting the urgent need for its conservation.

Given that *S. pedo* is a low-vagile [11] and parthenogenetic species [17], the preservation of open habitats and the maintenance of ecological connectivity between them is the only way to ensure the survival of relict populations and the preservation of their genetic variability [18]. For this reason, ecosystem planning and management should explicitly include connectivity assessments, identifying the most suitable sites for the maintenance of habitat connectivity in fragmented landscapes [19]. Thus, our aims are (i) to identify habitat requirements for *S. pedo* in the Northern Apennines by developing species distribution models (SDMs) and (ii) to identify ecological corridors for this species using the inverse of the resulting probability of occurrence map as a resistance surface through all cells of our study area.

## 2. Materials and Methods

### 2.1. Study Area

We carried out this study in a hilly region of about 2510 km^2^ in the Province of Alessandria (Figure 1). Our study area is characterized by a heterogeneous, fragmented landscape, featuring significant badlands formations and hosting a variety of habitats and species of Community interest, as defined by the EU Habitat Directive. These are protected within the Site of Community Importance (SCI) IT1180030 “Calanchi di Rigoroso, Sottovalle e Carrosio” [20]. These badlands are considered of landscape interest due to their distinctive geomorphology and ecological value. Broad-leaved forests dominate the environment, covering about half of the land [21]. The primary forest types include mesoxerophilous oak forests (*Quercus pubescens*), hop-hornbeam and manna ash forests (*Ostrya carpinifolia*, *Fraxinus ornus*), many of which are adept at colonizing upon grasslands, and chestnut groves (*Castanea sativa*). Other notable formations include broom shrublands (*Spartium junceum*) and xeric meadows. These habitats are under threat due to forest expansion, a consequence of the abandonment of mowing practices. This encroachment is also affecting former agricultural lands, which were previously maintained as improved grasslands but are now largely abandoned and transitioning into shrubland [21]. Within these open areas, dry and xeric meadows, largely attributable to habitat types (NATURA 2000 Code) 6210 and 6210* (important orchid sites), have been identified. These habitats are of Community interest due to the abundance of orchids, including rare species [20], and host our target species *S. pedo* [22].

### 2.2. Study Species and Data

We considered a total of 34 *S. pedo* occurrences (9 from Repetto et al. [23] + 25 occurrences of *S. pedo* collected in our study area by eight observers and stored on the platform iNaturalist, www.inaturalist.org, accessed on 10 February 2025). iNaturalist is an open-access platform designed for mapping and sharing biodiversity observations globally. It allows users to download species occurrences using specific queries (e.g., taxon, place, user/observer, date, etc.) [24]. Thus, we downloaded all *S. pedo* locations (with geographic coordinates) collected between April and July from 2018 to 2024, as this period corresponds to the highest biological activity and detection probability for our target species [24], between 44.4° N and 45° N of latitude and between 8.5° E and 9.5° E of longitude.

For the same time frame, we also gathered 1916 locations from four sites from Repetto et al. [24] in which our target species did not occur + 1912 pseudo-absence records of *S. pedo* reported between April and July, from 2018 to 2024, derived by other than our target species collected by the same eight observers of *S. pedo* to derive ‘observer-oriented’ (oo) pseudo-absences [25,26,27] from iNaturalist (including both plants and animals). We specifically used the ‘get_inat_obs’ and ‘get_inat_obs_user’ functions from the R package ‘rinat’ [28] to download *S. pedo* locations and those of other than our target species collected by the observers of *S. pedo*, respectively.

### 2.3. Predictor Variables

We initially evaluated nineteen predictors, including six topographic variables, ten land-cover variables, two forest structure variables, and one anthropogenic variable, to characterize the habitat of *S. pedo* (Table 1). The topographic variables were obtained from the TIN ITALY digital elevation model (DEM) with a 10 m spatial resolution (https://tinitaly.pi.ingv.it/Download_Area1_1.html, accessed on 15 August 2024). Land-cover features were sourced from the Copernicus CLC+ Backbone 2018, also with a 10 m spatial resolution (https://land.copernicus.eu/en/products/clc-backbone, accessed on 15 August 2024), and the Copernicus High Resolution Tree Cover Layer 2018, with the same spatial resolution (https://land.copernicus.eu/en/products/high-resolution-layer-tree-cover-density/tree-cover-density-2018, accessed on 15 August 2024). Among the land-cover features selected, grasslands refer to permanent herbaceous areas characterized by a continuous vegetation cover throughout a year (no bare soil occurs within a year) [29]. In our study, we evaluated both grasslands without woody trees (including mainly extensively managed natural grasslands or permanently managed grasslands, or arable areas with a permanent vegetation cover, e.g., fodder crops or even set-aside land in agriculture) and grasslands with few or many woody trees (woody trees ranging between 10 and 30% and 30 and 50%, respectively), mainly unmanaged [29]. Anthropogenic variables were derived from the Copernicus CLC+ Backbone 2018. All predictors were resampled to a 100 m spatial resolution.

Since multicollinearity (i.e., correlation among predictors) could dramatically bias the results of species distribution models (SDMs), we estimated a widely used index, the Variance Inflation Factor (VIF; [30]), for all the predictors considered. Predictors with VIF values > 3 were removed from the further analyses because of high multicollinearity among other predictors [30]. Technically, we used the ‘vifstep’ function in the R package 4.4.1 ‘usdm’ [31].

### 2.4. Data Analysis

To analyze the relationship between *S. pedo* occurrence and the selected predictors, we used the Integrated Nested Laplace Approximation, INLA [32], to regress occurrence and oo-pseudo-absence locations against predictor variables. The INLA provides a versatile modeling framework that can incorporate spatial random effects, i.e., the spatial dependency of species locations among each other, into binomial models, making it effective for generating predictions of spatial SDMs, i.e., avoiding considering species locations independent from each other in SDMs when actually they are not independent [33]. In this study, we developed a binomial model in the INLA, using *Saga pedo* presence/oo-pseudo-absence as the response variable, uncorrelated predictor variables as fixed effects, and spatial dependency among species locations through the Stochastic Partial Differential Equation (SPDE) approach [34], which relies on computations using a Gaussian Markov Random Field representation of the Gaussian Field [35]. Rather than fitting the INLA with only linear relationships between *S. pedo* occurrence and predictor variables (similar to a generalized linear model, GLM), we allowed the INLA to include smoothing parameters to account for non-linear relationships between predictors and the response variable (similar to a generalized additive model, GAM). Consequently, we combined INLA models with linear predictors (referred to as GLM-INLA) and non-linear predictors (referred to as GAM-INLA) into an ensemble prediction (wEP), weighted by the True Skill Statistic (TSS, see below). Here, we specify that, because our dataset consists of more species locations (and observer-oriented pseudo-absences) than the minimum identified by Erickson and Smith (2023) [36], we did not carry out SDMs specifically designed for rare species (e.g., ensemble of small models; Breiner et al., 2015 [37]).

To assess the predictive accuracy of our INLA models, we performed 10-fold cross-validation by using a random subsample of 90% of the locations for model calibration and the remaining 10% for evaluation. We used two commonly applied indices to evaluate model performance: (i) the area under the receiver operating characteristic curve (AUC) and (ii) the True Skill Statistic (TSS). The AUC ranges from 0 to 1, with 0 indicating a model worse than random and 1 indicating the best discriminating model. The TSS ranges from −1 to 1, where higher values indicate good predictive accuracy and 0 indicates random prediction.

We then converted the resulting continuous maps into binary maps, using threshold values estimated by maximizing the TSS [38], through the ‘bm_FindOptimStat’ function in the R package ‘biomod2’ [39]. Values higher and lower than these thresholds represented sites where *S pedo* was likely to occur and not likely to occur, respectively.

### 2.5. Landscape Connectivity Analysis

We derived a resistance-to-movement map as the inverse (1—probability of occurrence) of the resulting map from the INLA and used it as input in Omniscape.jl [40]. We choose Omniscape.jl over other approaches because it models landscape connectivity via random walks across all available movement possibilities. Similar to Circuitscape [41], Omniscape.jl relies on circuit theory and uses a ‘source-strength’ raster layer to ‘emit’ current, allowing each raster cell to reflect the relative abundance of potential dispersers. We specified the block size option as 1, so that every pixel of the resistance raster would be a target pixel. Omniscape.jl iteratively applied the Circuitscape ‘advanced’ mode algorithm, with a moving window with a specific radius of 300 m based on species dispersal ability [11,12,42], which provides a more accurate estimation of functional connectivity compared to the unrestricted current flow in Circuitscape [43].

## 3. Results

We found seven predictors with VIF values of >3 (multi-correlated; Table 1), and thus we considered the remaining twelve predictors in the further analyses.

Considering these remaining predictors, we estimated that 66.83 km^2^ (2.66%) of our study area was potentially suitable for *S. pedo* (Figure 2).

The probability of occurrence of *S. pedo* increased till it reached a plateau at 1000 m a.s.l., at a slope of 10°, while it decreased as human settlements and croplands increased, reaching the minimum between 50% and 75% in both the land-cover types (Figure 3). Shrublands, sparsely vegetated areas, and rocky areas shared a similar relationship with the probability of occurrence, though this pattern was more pronounced for shrublands and rocky areas; all these three land-cover types showed a high probability of occurrence around 50% for shrublands and sparsely vegetated areas, while it was closer to 40% for rocky areas (Figure 3). The probability of occurrence of *S. pedo* was not related to grasslands without woody trees (Figure 3). Conversely, the probability of occurrence of *S. pedo* was positively related to the coverage of both grasslands with few and with many trees (Figure 3). Finally, the probability of occurrence of *S. pedo* was slightly positively related to the percentage of water bodies and slightly negatively related with the Shannon habitat diversity index (Figure 3).

Ten-fold cross-validations showed the high predictive accuracy of both the GLM- and GAM-INLA SPDE (AUC: 0.939 ± 0.018 and 0.931 ± 0.021, respectively; TSS: 0.892 ± 0.045 and 0.904 ± 0.058, respectively) as well as those of their wEP (AUC and TSS: 0.918 ± 0.064 and 0.909 ± 0.052, respectively).

Landscape connectivity showed similar patterns of relationships to those of probability of occurrence with the considered predictor variables, except for shrublands, sparsely vegetated areas, rocky areas, and Shannon habitat diversity index (Figure 4). Landscape connectivity increased and reached a plateau close to 500 m a.s.l. and peaked close to a 5° slope (Figure 4). Human settlements, croplands, and grasslands without trees decreased to 25%, where they reached an asymptote (Figure 4). Conversely, shrublands were not related to landscape connectivity, while grasslands with few and many trees, over 25%, were positively related with landscape connectivity, as were sparsely vegetated areas, but only over 50%. Finally, landscape connectivity was negatively related to rocky areas and, to a lesser extent, to water bodies and Shannon habitat diversity index (Figure 4).

## 4. Discussion

Our study is set in a historical context where climate change and the evolution of land use in the Mediterranean region are seriously affecting open habitats and their affiliated species [44,45]. Moreover, the abandonment of traditional agropastoral activities, like itinerant pastoralism [46], now results in habitat loss due to afforestation [47,48,49]. Here, if on the one hand the endangered bush cricket *S. pedo* could benefit from rising temperatures caused by climate change [50], on the other, its expansion is severely limited by high fragmentation and isolation of xerothermic grasslands, making its potential expansion into new suitable territories difficult.

In our study, the most recent and robust species distribution and landscape connectivity modeling techniques were applied. We found that the occurrence and connectivity of *S. pedo* are hindered by intensive cultivation and favored by open habitats with woody trees, a condition that demands sustainable land management by humans.

### 4.1. Probability of Occurrence and Landscape Connectivity of S. pedo

Our findings indicate that *S. pedo* shows a significant preference for xerothermic grasslands and areas with a moderate abundance of woody trees. This preference aligns with its thermophilic and heliophilic characteristics, which favor warm, open habitats. The estimated suitable areas, which constitute 2.66% of our study area, are small and fragmented, posing a significant threat to the survival of the studied population of *S.pedo*. This threat is due to the species’ limited dispersal [11,12] and the increased risk of habitat loss from dense afforestation or intensive land use [24,51,52]. In our study area, the correlation between the occurrence of the bush cricket and altitude can further indicate its avoidance of intensive agriculture. Although *S. pedo* can occur in lowland areas [53], in our study area, this corresponds to the Po Valley, which is characterized by intensive agricultural practices [54] and urbanization [55]. Thus, we hypothesize that the preference for higher altitude identified in our study reflects a preference for natural and semi-natural areas, as demonstrated by Zema et al. (2022) [54], and represents the potential or actual distribution range of the species. The species’ preference for areas with some rocky or bare soil cover aligns with previously known empirical observations and is consistent with its reproductive ecology [22,52,53]. These substrates may provide suitable microhabitats for egg-laying, a key factor influencing the species’ habitat selection [53].

Furthermore, our analysis of landscape connectivity reveals the significant impact of intensive land use on the habitat connectivity of *S. pedo*. The expansion of croplands and the abandonment of traditional agricultural practices have led to a noticeable reduction in both the quantity and connectivity of suitable areas. This ongoing decline in suitable habitats and connectivity poses a serious threat to the population’s survival as reduced connectivity limits the species’ ability to colonize new suitable habitats, disrupts prey flows with similar ecological requirements, and increases the extinction risk of small local subpopulations [18,56,57]. Our findings support a previous study on the impact of agriculture on connectivity [58], highlighting the fragility of corridors for this species and underscoring the urgent need for conservation or intervention.

### 4.2. INLA-SPDE and Omniscape.jl for the Conservation of S. pedo

To strengthen conservation efforts, we developed SDMs using ensemble predictions from INLA-SPDE, thus reducing single model uncertainty [59] while accounting for spatial autocorrelation, often neglected in traditional SDMs [60], thereby providing more accurate predictions of species distribution. Moreover, we used the inverse of occurrence probability from INLA-SPDE as a resistance map for Omniscape.jl, rather than relying on subjective expert-based resistance maps, ensuring a data-driven and objective assessment of landscape connectivity [61]. Omniscape.jl, the ‘evolution’ of Circuitscape—still widely used worldwide [62,63,64] and already tested on threatened insect species [65]—offers advanced capabilities for modeling omni-directional landscape connectivity, making it a crucial tool for effective conservation planning.

Thus, by integrating INLA-SPDE with Omniscape.jl, we are confident that we can better understand and mitigate the threats to *S. pedo*, ensuring its long-term survival and ecological role.

## 5. Conclusions

Our findings highlight the urgent need for integrated conservation strategies that address both habitat preservation and landscape connectivity, emphasizing the importance of existing suitable habitats for the survival of *S. pedo* populations. Moreover, in order for the species to persist in its territory, it is essential to improve ecological corridors and to mitigate the impacts of land use changes for maintaining connectivity across fragmented landscapes.

Land management practices can contribute to the long-term survival of this vulnerable species by promoting sustainable agricultural activities and by managing abandoned meadows and pastures so as to contain the advance of forest vegetation. Additionally, in order to maintain the integrity of the species’ critical habitats, restoring degraded environments and expanding protected areas is necessary.

Further studies examining the species’ responses to climate and land use change scenarios can provide deeper insights for developing *S. pedo* conservation strategies. Furthermore, expanding research to other regions may help generalize our findings and refine conservation approaches for *S. pedo* and many other species with similar ecological needs.

## Figures and Tables

**Figure 1 insects-16-00279-f001:**
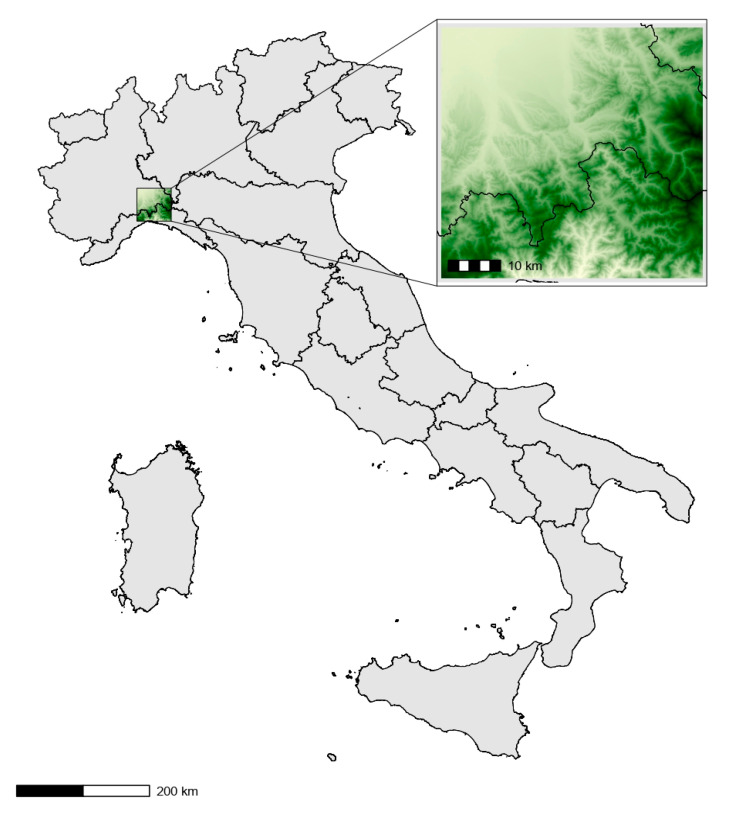
Study area. Black lines indicate Italian regional borders. Light–dark green scale indicates lower–higher elevation.

**Figure 2 insects-16-00279-f002:**
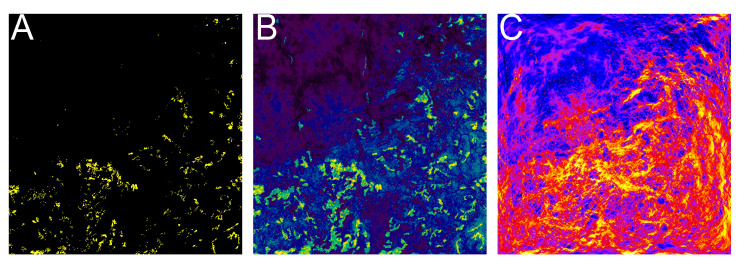
Distribution maps of *Saga pedo* estimated by weighted ensemble prediction of GLM- and GAM-INLA SPDE and landscape connectivity with Omniscape.jl. (**A**) Areas of predicted species occurrence estimated using a threshold value of 64.01 (threshold values estimated by maximizing TSS): presence indicated by yellow; absence indicated by black. (**B**) Probability of occurrence: yellow–blue scale indicates higher–lower occurrence probability values, respectively. (**C**) Landscape connectivity: yellow–blue scale indicates higher–lower landscape connectivity values, respectively.

**Figure 3 insects-16-00279-f003:**
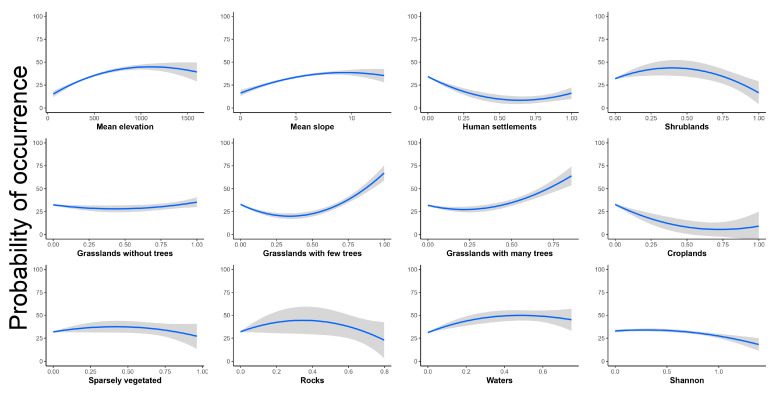
Response curves (in blue) and relative 95% confidence intervals (in gray) of probability of occurrence of *Saga pedo* in relation to predictor variables.

**Figure 4 insects-16-00279-f004:**
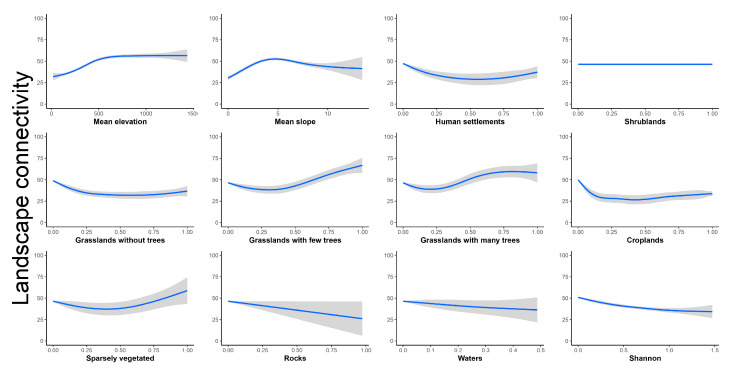
Response curves (in blue) and relative 95% confidence intervals (in gray) of landscape connectivity of *Saga pedo* in relation to predictor variables.

**Table 1 insects-16-00279-t001:** Variables used in the development of GLM- and GAM-INLA models for *Saga pedo*. Variables with Variance Inflation Factor (VIF) > 3 were removed due to multicollinearity with other variables.

Predictor	Unit	VIF
Mean elevation	m a.s.l.	1.507
Stand. dev. elevation	m a.s.l.	>3
Mean slope	°	1.907
Stand. dev. slope	°	>3
Mean roughness	average length of isoipses in the cell/cell side	>3
Stand. dev. roughness	average length of isoipses in the cell/cell side	>3
Mean tree cover density	n/m^2^	>3
Stand. dev. tree cover density	n/m^2^	>3
Shrublands	%	1.117
Grasslands without woody trees	%	1.338
Grasslands with few woody trees	%	1.238
Grasslands with many woody trees	%	1.197
Croplands	%	2.388
Sparsely vegetated areas	%	1.048
Waters	%	1.055
Woodlands	%	>3
Rocky areas	%	1.041
Shannon habitat diversity index	H′ = −Σ (pi × lnpi)	1.966
Human settlements	%	1.231

## Data Availability

The original contributions presented in this study are included in the article. Further inquiries can be directed to the corresponding author.

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
