# Peer review of "Identifying Ecological Corridors of the Bush Cricket Saga pedo in Fragmented Landscapes"

_insects, 2025, doi:10.3390/insects16030279_

Round 1
Reviewer 1 Report
Comments and Suggestions for Authors
This paper presents an interesting framework to model the local distribution of a protected insect, also exploring the parameters that drive habitat occurrence and connectivity within a landscape mosaic. The manuscript is clear, concise and well presented. However, I feel there are a couple of major issues regarding methodological choices that, in turn, could reflect some misleading interpretation of the results.
While SDM gives the opportunity to predict habitat distribution and connectivity, some results of the INLA models sound misleading, or at least difficult to interpret. As a reader, I have the feeling that there are two levels of issues that need to be further addressed in the manuscript:
Among the drivers of species occurrence and connectivity, it is somehow strange that grasslands without trees had such a negative effect on species occurrence probability, similarly to croplands and woodlands. Based on the information provided, I could not exclude that this effect is due to some bias with habitat interpretation or the spatial resolution, depending on the quality of the CLC maps and/or the heterogeneity of the resampled pixels. Moreover, since you report that grasslands without trees also include cultivated fallow fields and fodder crops (which can hardly be considered grasslands from the point of view of a grassland-specialist insect), I feel that the positive effects of grasslands with trees only rely on a less-intensive management of their herbaceous layer, rather than on the tree cover itself. I would suggest that you verify these perceptions (e.g. by visually interpreting recent orthophotographs) and, in any case, that you provide more explanation on the possible interpretations and limitations of this result in the Discussion section.
Since Saga pedo have a low dispersal ability, and it is already known to be strictly depending on patches of open semi-natural vegetation, I cannot clearly see the reason for testing the distance from target habitats as a predictor of species distribution and connectivity. From the point of view of a non-flying and weakly-moving insect, I would hypothesize a very important role of local habitat availability, but a relatively little importance of the surrounding context and proximity to further habitat sources. Moreover, an eventually questionable selection of predictors may also be relevant to results of your study, considering that adding more risk of multicollinearity affect the final variables’ selection. Please, consider revising the list of tested variables, based on a more clear meaning for species ecological requirements. Otherwise, please provide a clear explication of the reasons behind these choices, which could also further support the interpretation of results.
Here below, few specific comments:
Title: Based on the title, I would expect a study oriented at identifying fine-scale corridors within a given habitat type. Actually, also due to the considered scale, the proposed framework does not seem to be able to identify corridors within a specific vegetation type, but rather at landscape level. I would cut the last part of the title, or maybe consider saying “… in fragmented landscapes”. Anyhow, please check the correct word “semi-natural grasslands” rather than “seminatural-grasslands” More in general, I feel that the study did not stress very much on the function and role of corridors, but rather on the predictors affecting their conservation in general. I would suggest more brainstorming to choose a title that could better help framing the study.
In general, please consider breaking long sentences to improve their readability (e.g., lines 46-48).
Lines 52-54: I would not say that its ecological preferences would make this species more sensitive to habitat fragmentation, but rather, this can be due to other specific traits (e.g. its low mobility) and/or supported by studies demonstrating that a sufficient size/connectivity of habitat patches should be required. I would suggest introducing a bit more about the potential needs and constraints for Saga pedo, based on its ecology and in terms of connectivity, as it is the main objective of the study.
Line 54: “their fragmentation” instead of “its fragmentation”.
Line 64: here you refer to “this relict population and the preservation of its genetic variability”, but there is no mention of the target population in the preceding paragraphs. You can either rephrase this sentence or tell something more about the specific case of the Northern Apennines’ population.
Line 181 – I would start the paragraph with “From previous work” or “From Repetto et al.” rather than the bracketed citation alone.
Line 182 – replace “did not occurred” with “did not occur”.
Author Response
Comment 1: Since Saga pedo have a low dispersal ability, and it is already known to be strictly depending on patches of open semi-natural vegetation, I cannot clearly see the reason for testing the distance from target habitats as a predictor of species distribution and connectivity. From the point of view of a non-flying and weakly-moving insect, I would hypothesize a very important role of local habitat availability, but a relatively little importance of the surrounding context and proximity to further habitat sources. Moreover, an eventually questionable selection of predictors may also be relevant to results of your study, considering that adding more risk of multicollinearity affect the final variables’ selection. Please, consider revising the list of tested variables, based on a more clear meaning for species ecological requirements. Otherwise, please provide a clear explication of the reasons behind these choices, which could also further support the interpretation of results.
Response 1: We understand your concern and have removed all distance-related predictors initially considered. We have also recalculated multicollinearity among the remaining variables.
Comment 2: Title: Based on the title, I would expect a study oriented at identifying fine-scale corridors within a given habitat type. Actually, also due to the considered scale, the proposed framework does not seem to be able to identify corridors within a specific vegetation type, but rather at landscape level. I would cut the last part of the title, or maybe consider saying “… in fragmented landscapes”. Anyhow, please check the correct word “semi-natural grasslands” rather than “seminatural-grasslands” More in general, I feel that the study did not stress very much on the function and role of corridors, but rather on the predictors affecting their conservation in general. I would suggest more brainstorming to choose a title that could better help framing the study.
Response 2: We understood and agreed with this suggestion and thus modified the title accordingly: Identifying ecological corridors of the bush cricket Saga pedo in fragmented landscapes
Comment 3: In general, please consider breaking long sentences to improve their readability (e.g., lines 46-48).
Response 3: Corrected.
Comment 4: Lines 52-54: I would not say that its ecological preferences would make this species more sensitive to habitat fragmentation, but rather, this can be due to other specific traits (e.g. its low mobility) and/or supported by studies demonstrating that a sufficient size/connectivity of habitat patches should be required. I would suggest introducing a bit more about the potential needs and constraints for Saga pedo, based on its ecology and in terms of connectivity, as it is the main objective of the study.
Response 4: We have revised the paragraph (Lines 52-54) to clarify that Saga pedo's sensitivity to habitat fragmentation is primarily due to its limited mobility rather than solely its ecological preferences. The revised text now explicitly highlights this constraint and the importance of habitat connectivity for its conservation.
Comment 5: Line 54: “their fragmentation” instead of “its fragmentation”.
Response 5: corrected.
Comment 6: Line 64: here you refer to “this relict population and the preservation of its genetic variability”, but there is no mention of the target population in the preceding paragraphs. You can either rephrase this sentence or tell something more about the specific case of the Northern Apennines’ population.
Response 6: We have rephrased the sentence in Line 64 to improve clarity and avoid the lack of context.
Comment 7: Line 181 – I would start the paragraph with “From previous work” or “From Repetto et al.” rather than the bracketed citation alone.
Response 7: corrected
Comment 8:Line 182 – replace “did not occurred” with “did not occur”.
Response 8: corrected.
Reviewer 2 Report
Comments and Suggestions for Authors
This paper by Della Rocca, Repetto, De Caria, and Milanesi is a valuable contribution to insect conservation ecology. The authors use current methods to create a SDM for the IUCN threatened katydid Saga pedo. The study also illustrates a powerful potential use of the community science platform iNaturalist. The paper is concise and the content is generally clear.
General comments:
My main methodological question centers around the pseudo-absence data. This is a contentious topic in GIS and SDM in particular. The authors derived pseudo-absence data from geographic user samples of non-target species in suitable habitat for S. pedo, implying that S. pedo was absent. This is one of two common choices, the other being random background sampling (Zbinden et al. 2024, Ecological Informatics 81). One of the authors, P. Milanesi, published on this approach and shows improvements over random sampling. I would nonetheless include more in the Methods about how the authors decided to mine and use pseudo-absence data. Does S. pedo have a community of species that it frequently co-occurs with that are prime choices for pseudo-absence points? Even better if there are other grassland Orthoptera that co-occur with S. pedo, or common prey. Mentioning the presence of such species lends weight to the chosen pseudo-absence data. The authors do not mention the non-target species that were chosen, but do provide a quantity, the size of which suggests it is drawn from an entire biota, which is in line with the reference (Milanesi et al 2020 Ecology and Evolution 10). Citizen scientists sample for iNaturalist out of a variety of goals: some will be looking for particular species, often rare or sought-after ones. Other users inventory habitats for particular taxa. The range of user/observer behavior and motivation is may thus bias the use of their data for pseudo-absence purposes. Saga are large and striking insects but also cryptic in grass and may be overlooked by untrained or uninterested observers. The number of observers on iNaturalist was reported at 7, not a high number, so bias will count more for this study.
A second question has to do with the historical ecology of Saga pedo. This species is currently widespread but uncommon throughout southern Europe. The authors may wish to explain the recent history of the type of forested habitats that they identified in Europe before extensive human modification. Are savanna, parkland, or sparsely wooded grassland something that was widespread in Europe in contiguous patches of large area, or is this habitat type naturally fragmented historically? Has this species perhaps always existed in habitat fragments, and thus human-driven habitat fragmentation will less severely affect this species? This is particularly interesting given the mention of conflicting outcomes in the Discussion: deforestation and increased aridity with climate change on one hand, land use changes and habitat fragmentation on the other. Is Saga pedo a species that one has a reasonable chance of finding in suitable habitat, or are there large areas of suitable habitat that the species is missing from?
Specific comments:
Methods
P3 L77. Explain what “landscape interest” is. Landscapes relevant to this study, where iNaturalist observations were abundant, or areas where conservation will eventually be implemented? A combination of these?
P3 L86. Similar to above, explain “community interest.” If rare species are present, the areas are of conservation interest, and would therefore also interest a segment of the iNaturalist user population. This statement could also mean areas of interest to the study organism, the ecological community to which it belongs.
Table 1. Shannon habitat diversity index line. Use pi if possible, so as not to be confused with mathematical pi.
Results
P6 L181 First sentence has missing item(s).
P6 L181-183 The sentence is difficult to understand, please rephrase.
P6 L186 Single sentence cannot be a paragraph. Combine with the previous or following paragraph.
P6 L186-187 Unclear, consider changing to a statement like "The pool of potential predictor variables was reduced from 29 to 12 by screening VIF values less than or equal to 3."
Discussion
P8 L244 Word choice "implied." I read this sentence to mean that the results of this study sustainable land management is required or prompted or demanded of humans, one of those words may be better.
Author Response
General comments:
Comment 1: My main methodological question centers around the pseudo-absence data. This is a contentious topic in GIS and SDM in particular. The authors derived pseudo-absence data from geographic user samples of non-target species in suitable habitat for S. pedo, implying that S. pedo was absent. This is one of two common choices, the other being random background sampling (Zbinden et al. 2024, Ecological Informatics 81). One of the authors, P. Milanesi, published on this approach and shows improvements over random sampling. I would nonetheless include more in the Methods about how the authors decided to mine and use pseudo-absence data. Does S. pedo have a community of species that it frequently co-occurs with that are prime choices for pseudo-absence points? Even better if there are other grassland Orthoptera that co-occur with S. pedo, or common prey. Mentioning the presence of such species lends weight to the chosen pseudo-absence data. The authors do not mention the non-target species that were chosen, but do provide a quantity, the size of which suggests it is drawn from an entire biota, which is in line with the reference (Milanesi et al 2020 Ecology and Evolution 10). Citizen scientists sample for iNaturalist out of a variety of goals: some will be looking for particular species, often rare or sought-after ones. Other users inventory habitats for particular taxa. The range of user/observer behavior and motivation is may thus bias the use of their data for pseudo-absence purposes. Saga are large and striking insects but also cryptic in grass and may be overlooked by untrained or uninterested observers. The number of observers on iNaturalist was reported at 7, not a high number, so bias will count more for this study.
Response 1: We agree with this Reviewer about the different behaviour of citizen scientists, but, while elusive, given the rarity of S. pedo, we expect that the (few) observers of our target species would have recorded its occurrence in case of an observation and thus expect that where they collected other than the target species occurrence, they did not observe any S. pedo. However, we specified in the discussion section, as possible improvements of our approach, that researchers could additionally refine the list of pseudo-absences based on previously published papers on S. pedo co-occurring species.
Comment 2: A second question has to do with the historical ecology of Saga pedo. This species is currently widespread but uncommon throughout southern Europe. The authors may wish to explain the recent history of the type of forested habitats that they identified in Europe before extensive human modification. Are savanna, parkland, or sparsely wooded grassland something that was widespread in Europe in contiguous patches of large area, or is this habitat type naturally fragmented historically? Has this species perhaps always existed in habitat fragments, and thus human-driven habitat fragmentation will less severely affect this species? This is particularly interesting given the mention of conflicting outcomes in the Discussion: deforestation and increased aridity with climate change on one hand, land use changes and habitat fragmentation on the other. Is Saga pedo a species that one has a reasonable chance of finding in suitable habitat, or are there large areas of suitable habitat that the species is missing from?
Response 2: Thank you for your insightful comment. The observation regarding the historical landscape before extensive human modification is particularly relevant. However, in the Mediterranean context, human influence on the landscape dates back millennia, making it difficult to define a truly "natural" pre-modification state.
As we mention in our manuscript (lines 46-49), " Nonetheless, semi-natural grasslands in Europe are experiencing a significant decline and fragmentation [3]. This is largely due to their conversion into high-yield crops or the cessation of farming and pastoral activities, such as mowing and extensive grazing. These changes have, in turn, favored the expansion of forest [4].” Additionally, following the extinction of large herbivores in Europe, the persistence of grassland habitats and their biodiversity has mostly depended on moderate disturbance by human activities, primarily through grazing and mowing (Pärtel et al., 2005, https://lucris.lub.lu.se/ws/portalfiles/portal/5528474/625284.pdf).
Given this, Saga pedo may have historically existed in naturally fragmented habitats, which could suggest some resilience to human-driven habitat changes. However, the ongoing decline of semi-natural grasslands raises concerns about habitat availability and connectivity, potentially influencing the species’ distribution. Understanding how these landscape changes interact with Saga pedo’s ecology remains crucial, particularly in light of the contrasting effects of climate change and land use shifts discussed in our study.
Specific comments:
Methods
Comment 3: P3 L77. Explain what “landscape interest” is. Landscapes relevant to this study, where iNaturalist observations were abundant, or areas where conservation will eventually be implemented? A combination of these?
Response 3: We have clarified “landscape interest” by specifying that the badlands are valued for their geomorphology and ecological significance.
Comment 4: P3 L86. Similar to above, explain “community interest.” If rare species are present, the areas are of conservation interest, and would therefore also interest a segment of the iNaturalist user population. This statement could also mean areas of interest to the study organism, the ecological community to which it belongs.
Response 4: we have now clarified what with mean with “Community interest” at line 83
Comment 5: Table 1. Shannon habitat diversity index line. Use pi if possible, so as not to be confused with mathematical pi.
Response 5: We understood and agreed with this suggestion and thus modified accordingly.
Results
Comment 6: P6 L181 First sentence has missing item(s).
Response 6: Corrected.
Comment 7: P6 L181-183 The sentence is difficult to understand, please rephrase.
Response 7: We agree, rephrased.
Comment 8: P6 L186 Single sentence cannot be a paragraph. Combine with the previous or following paragraph.
Response 8: Corrected.
Comment 9: P6 L186-187 Unclear, consider changing to a statement like "The pool of potential predictor variables was reduced from 29 to 12 by screening VIF values less than or equal to 3."
Response 9: We agree, changed as you suggested.
Comment 10: P8 L244 Word choice "implied." I read this sentence to mean that the results of this study sustainable land management is required or prompted or demanded of humans, one of those words may be better.
Response 10: Changed with “demands”.
Reviewer 3 Report
Comments and Suggestions for Authors
General comments.
The work presented by Della Rocca et al. is based on a modelling approach to identify among 29 habitats variables which one are the best predictors for S. pedo occurrence and identify ecological corridors for this species. This mathematical approach is quite difficult to understand for persons (most entomologists and conservationists?) who lack being familiar with modelling, including me.
For example:
“To prevent multicollinearity among predictors from affecting the species distribution 125 models (SDMs), we calculated the Variance Inflation Factor (VIF; [25] for all predictor val-126 ues throughout our study area. Using the ‘vifstep’ function in the R package ‘usdm’ [26] we performed a stepwise selection analysis, removing variables until the highest VIF value was less than…”
And in data analysis section:
“To analyze the relationship between S. pedo occurrence and the selected predictors, we used the Integrated Nested Laplace Approximation, INLA [27] to regress occurrence and oo-pseudo-absence locations against predictor variables. INLA provides a versatile modelling framework that can incorporate spatial random effects into binomial models, making it effective for generating SDM-type spatial predictions [28]. In this study, we developed a binomial model in INLA, using Sp presence/oo-pseudo-absence as the response variable, uncorrelated predictor variables as fixed effects, and spatial dependency among species locations through the Stochastic Partial Differential Equation (SPDE) approach [29]…”
As I am not myself familiar with modelling, I can’t comment the method and I have no other option but to trust the authors. I recommended the editor to take care that the second reviewer will have the necessary mathematical skills. So my comments will focus on the ecological and entomological/conservation value of this work.
First, I have to say that the main results of this work is to mathematically confirm something that was already empirically known by field entomologists. Della Rocca et al. write “Our results showed that S. pedo prefers xerothermic grasslands with moderate woody vegetation and avoids areas characterized by intensive agriculture” which echoes the findings of Kritin and Kanuch (2007) “This species is recorded in Slovakia in xerothermic forest steppes and limestone grikes (98% of localities)”, the assumption of Holuša et al. (2013) “Saga pedo is an important xerothermophilous and pratinicolous species that indicates original forest–steppe biotopes (Kaltenbach 1970; Kristın and Kanuch 2007; Lemonnier-Darcemont et al. 2009)” or the species habitat description provided at http://www.pyrgus.de/Saga_pedo_en.html. “Saga pedo inhabits dry, warm, richly structured grassland slopes and maquis with bushes and higher growing areas from the lowlands to about 1400m above sea level”.
Second, I do not understand the choice made by Della Rocca et al. to focus their writing on the concepts of “connectivity” and “ecological corridors”. Indeed, Della Rocca et al. “derived a resistance to movement map as the inverse (1 – probability of occurrence)” but this mathematical formula corresponds to the definition of an “habitat unsuitability” index rather than to a real “resistance to movement” index. Furthermore, movements and even more dispersion movement of S. pedo are very poorly known. Due to the difficulties to find S. pedo specimens, only Richard (2010, master thesis), Holuša et al. (2013), Chesné (2014, master thesis) and Anselmo (2022) intended capture/mark/recapture studies to estimate some movement parameters.
Based on the monitoring of 15 adults, Richard (2010) showed that S. pedo moved an average distance of 2,87 m per day and an average total distance of 38,82m during, in average, 15,67 days.
Holuša et al. (2013) monitored both nymph and adult movements. They wrote “Based on the recapture of marked adults at the Novy´hrad and Martinka sites, adult S. pedo (n = 16) moved between 0.5 and 2 m in 24 h. One female moved 14.0 m and another 37.5 m during 2006 and 2007, respectively”
Chesné (2014) marked and monitored 8 females and provide unclear results: hourly movements of 0 up to 2,857 m per hour.
Anselmo 2022 monitored 7 individuals during 33 days and estimated a “mean daily shift of 2.9 m”.
Based on this literature, it is unclear how Della Rocca et al. decided to use a “moving window with a specific radius of 300m based on species dispersal ability”. None of those available master thesis or peer-reviewed article clearly allow to estimate the total distance travelled during the lifespan of a S. pedo female due to the lack of directional information. Only Richard (2010) indicate that despite total distance travelled is 38,82m / 15,67 days in average, distance between the first and last observation was in average almost half shorter: 21,93 m.
We would rather advocate to focus on the conservation of suitable habitats per se as Nuhlíčková et al. (2024) do with the endemic flightless bush-cricket, Isophya beybienkoi.
Detailed comments.
Line 5 (authors affiliations): Insert coma and dot to obtain “Pavia, Italy.” (in a way to homogenise with the others addresses).
Line 8: use capital letters to obtain “Via” instead of “via”.
Line 45: Please consider that “delicate” is not an appropriate adjective here.
Line 46-47-48: I would suggest to shorten / divide this long sentence.
Line 51: I’m surprised the authors mention that “S. pedo” will refer to Saga pedo. This is a very common writing convention, not only for entomologists. I thus think “S. pedo, hereafter » should be deleted.
Line 52 (“lead it to concentrate in xerothermic patches”) and 54 (“one of the first to feel the effects of its fragmentation »): Those terms are not appropriate. S. pedo is surely sensitive to its habitat loss per se, but there is no published work indicating it is sensitive to its habitat fragmentation per se.
Line 56-57: Idem. There is no published work indicating “the reduction in ecological connectivity between the remaining open patches” participate in S. pedo decline.
Line 58-60: Authors should add that S. pedo was considered a least concern species in Europe (Hochkirch et al. 2016), should provide the year of publication of the European Union Habitats Directive and the year of evaluation of the global conservation status by the IUCN.
Line 64-69: I would not insist on the importance of “connectivity”, “connectivity assessments”, “maintenance of habitat connectivity”, “ecological corridors” for “the survival of this relict population and the preservation of its genetic variability”.
There is poor information about the movement of S. pedo nymphs and adult females and nothing is known about the genetic variability of S. pedo (but there is poor chance that this genetic variability plays a major role in the conservation of this exclusive parthenogenetic species).
Line 75: “in” is repeated twice. Delete one.
Line 76: Delete coma in “Alessandria, (Fig. 1)”.
Line 79-80: Add hornbeam-elm and chestnut trees latin names (as you did right before for Quercus pubescens).
Line 82: Replace “environments” by “habitats”.
Line 81-84: Please clarify “due to the cease of mowing, similar to abandoned agricultural and cultivated areas, which are mainly characterized by improved grasslands”. (I do not understand “similar to abandoned agricultural and cultivated areas” nor “improved grasslands”).
Figure 1: Scale bars should be added and the figure legend should contain more details. Please indicate that the right part is a map of Italy with administrative province / region limits. Please add some localities to be used as geographic markers.
Line 92: Please clearly indicate the number of S. pedo occurences obtained from Repetto et al. (2024) and the number of S. pedo occurences obtained from INaturalist.
Line 98: I agree that the period between April and July is the period of maximal biological activity, however, I’m surprised that the authors did not search to obtain as many data as possible to feed their mathematical models/analysis and choose to discard the data pool obtained in August-September. Furthermore, if I were to expect dispersal within a species, I would expect it lately in the breeding season (Lakovic et al. 2024).
Beside the time period, shouldn’t you provide details about the geographic perimeter of your query on INaturalist ?
Line 101: Please clarify what you mean by “… to derive ‘observer-oriented’ (oo) pseudo-ab-101 sences [20–22] from iNaturalist (including both plants and animals)”.
I guess that you assumed that your observers had always uploaded the presence of S. pedo in case they observed it. What if it they did not?
Shouldn’t you provide more details about your use of non-Saga pedo observations? I think I understand that you considered as an absence of S. pedo a location where the observers reported non-Saga pedo observations but did not report S. pedo. However, I wonder if you shouldn’t have used a threshold value. For example, at a location where less than “x” (x=5? 10? …) non target species were reported, the search effort could be considered as too low to expect someone to find S. pedo even if the species is present.
Line 112: Could you please explain what is “Copernicus CLC+ Backbone 2018”? Aren’t you supposed to provide the DOI for this source? It is provided in the acknowledgments section but it would be more appropriate here.
Line 118: Add a coma after “In our study”.
Line 125 and 132: Please choose how to spell “multicollinearity” / “multi-collinearity”
Line 135: the whole chapter “data analysis” is very difficult to understand for a person who is not used to species distribution models.
Line 141: Use “S. pedo” instead of Sp.
Line 181-185: This should be included in the “Mat&Meth” section rather than in “Results” section.
Line 183: You refer to figure 1 so the reader should expect to find the 23 S. pedo occurrences in your study site. However, actually there is no S. pedo occurrence shown in your figure 1.
Line 184: As there are only seven observers, I think the authors should cite them. Currently they are not even thanked by name in the acknowledgements section.
Line 188: Is “reaming predictors” correct? Shouldn’t be “remaining predictors”?
Line 196 and 198: Shouldn’t “the species occurrence” be replaced by “the S. pedo occurrence”?
Line 212, 213, 216: Replace “%”by “percentage”
Line 217: I would not use “connectivity” but “habitat suitability”.
Line 223: Shouldn’t “predicted species occurrence” be replaced by “predicted S. pedo occurrence”?
Figure 2. Authors should provide some geographic markers and scale bars within their maps to make them useful for local biodiversity managers / stakeholders. Currently those figures can’t be said to be “maps”.
I would have expected the “Landscape connectivity” to be the photographic negative of the “Probability of occurrence map” but it is not the case. Is it normal?
Line 235-236: This reference to the exploitation of natural resources is not appropriate. The authors should rather focus on the Mediterranean context evolution affecting open habitat and their affiliated species: the abandonment of itinerant pastoralism, the abandonment of agriculture in low productivity lands, first leading to an increase in habitats availability but now leading to its decrease due to afforestation.
Line 242-243. I would rather insist on the precise definition of relative weight of the different habitat suitability parameters.
Line 255-256: S. pedo do not only have the potential to live in lowlands: it lives in lowlands when possible (see altitudinal distribution of the species in Occitanie-France : https://biodiv-occitanie.fr//espece/65680)
Line 278: I agree with authors that mathematical models and their “data-driven” conclusion are useful. However, I’m not so confident in models when they are “fed” with only 23 species observations.
Line 283-285: One would have expected the authors to provide some clear practical guidelines for land managers and hierarchization rules: where connectivity should be restored precisely? And where connectivity restoration will provide the greatest benefits?
References list.
Please check references list and ensure species latin names (Saga pedo, Popillia japonica) are in italic. Use capital letters only for the first letter of the genus name, not for the species name.
Author Response
General comments.
Comment 1: This mathematical approach is quite difficult to understand for persons (most entomologists and conservationists?) who lack being familiar with modelling, including me.
For example:
“To prevent multicollinearity among predictors from affecting the species distribution 125 models (SDMs), we calculated the Variance Inflation Factor (VIF; [25] for all predictor val-126 ues throughout our study area. Using the ‘vifstep’ function in the R package ‘usdm’ [26] we performed a stepwise selection analysis, removing variables until the highest VIF value was less than…”
Response 1: We understood the concern of this Reviewer and thus modified the entire methods section in order to make it easier to read than the previous version also to readers non-familiars with these approaches.
Comment 2: And in data analysis section:
“To analyze the relationship between S. pedo occurrence and the selected predictors, we used the Integrated Nested Laplace Approximation, INLA [27] to regress occurrence and oo-pseudo-absence locations against predictor variables. INLA provides a versatile modelling framework that can incorporate spatial random effects into binomial models, making it effective for generating SDM-type spatial predictions [28]. In this study, we developed a binomial model in INLA, using Sp presence/oo-pseudo-absence as the response variable, uncorrelated predictor variables as fixed effects, and spatial dependency among species locations through the Stochastic Partial Differential Equation (SPDE) approach [29]…”
Response 2: We revised these sentences to clarify the methods, as it follows: … INLA provides a versatile modelling framework that can incorporate spatial random effects, i.e., the spatial dependency of species locations among each other, into binomial models, making it effective for generating predictions of spatial SDMs, i.e., avoiding considering species locations independent from each other in SDMs when actually they are not independent [28 ].
Comment 3: I do not understand the choice made by Della Rocca et al. to focus their writing on the concepts of “connectivity” and “ecological corridors”. Indeed, Della Rocca et al. “derived a resistance to movement map as the inverse (1 – probability of occurrence)” but this mathematical formula corresponds to the definition of an “habitat unsuitability” index rather than to a real “resistance to movement” index. Furthermore, movements and even more dispersion movement of S. pedo are very poorly known. Due to the difficulties to find S. pedo specimens, only Richard (2010, master thesis), Holuša et al. (2013), Chesné (2014, master thesis) and Anselmo (2022) intended capture/mark/recapture studies to estimate some movement parameters.
Based on the monitoring of 15 adults, Richard (2010) showed that S. pedo moved an average distance of 2,87 m per day and an average total distance of 38,82m during, in average, 15,67 days.
Holuša et al. (2013) monitored both nymph and adult movements. They wrote “Based on the recapture of marked adults at the Novy´hrad and Martinka sites, adult S. pedo (n = 16) moved between 0.5 and 2 m in 24 h. One female moved 14.0 m and another 37.5 m during 2006 and 2007, respectively”
Chesné (2014) marked and monitored 8 females and provide unclear results: hourly movements of 0 up to 2,857 m per hour.
Anselmo 2022 monitored 7 individuals during 33 days and estimated a “mean daily shift of 2.9 m”.
Based on this literature, it is unclear how Della Rocca et al. decided to use a “moving window with a specific radius of 300m based on species dispersal ability”. None of those available master thesis or peer-reviewed article clearly allow to estimate the total distance travelled during the lifespan of a S. pedo female due to the lack of directional information. Only Richard (2010) indicate that despite total distance travelled is 38,82m / 15,67 days in average, distance between the first and last observation was in average almost half shorter: 21,93 m.
We would rather advocate to focus on the conservation of suitable habitats per se as Nuhlíčková et al. (2024) do with the endemic flightless bush-cricket, Isophya beybienkoi.
Response 3: We greatly appreciate this Reviewer for providing additional references on the movement of Saga p., now included in this updated version of our manuscript. Initially, we used the mean daily distance of 2 m from Holuša et al. (2013) and multiplied it by the duration of the period of activity of S. pedo, from April to July (N = 122 days). This equals a total distance of 224 m, which is more than twice our cell size of 100 m. Therefore, we considered a moving window with a specific radius of 300 m in Omniscape.jl. However, taking into account the findings of Richards (2010), which report an average daily movement of 2.87 m or 38.82 m over 15.67 days, and Anselmo (2022), which indicates an average daily movement of 2.9 m, we derived an average movement distance of 307.48 m over 122 days from these studies (including also Holuša et al., 2013). This is very close to our moving window with a specific radius of 300 m in Omniscape.jl and thus we are now even more confident than before that our choice of a 300 m radius is consistent.
Detailed comments
Comment 4: Line 5 (authors affiliations): Insert coma and dot to obtain “Pavia, Italy.” (in a way to homogenise with the others addresses).
Response 4: corrected.
Comment 5: Line 8: use capital letters to obtain “Via” instead of “via”.
Response 5: corrected.
Comment 6: Line 45: Please consider that “delicate” is not an appropriate adjective here.
Response 6: corrected.
Comment 7: Line 46-47-48: I would suggest to shorten / divide this long sentence.
Response 7: corrected.
Comment 8: Line 51: I’m surprised the authors mention that “S. pedo” will refer to Saga pedo. This is a very common writing convention, not only for entomologists. I thus think “S. pedo, hereafter » should be deleted.
Response 8: corrected.
Comment 9: Line 52 (“lead it to concentrate in xerothermic patches”) and 54 (“one of the first to feel the effects of its fragmentation »): Those terms are not appropriate. S. pedo is surely sensitive to its habitat loss per se, but there is no published work indicating it is sensitive to its habitat fragmentation per se.
Comment 10: Line 56-57: Idem. There is no published work indicating “the reduction in ecological connectivity between the remaining open patches” participate in S. pedo decline.
Response 9 &10: We have revised the text to ensure accuracy in describing Saga pedo's sensitivity to habitat changes. Specifically, we no longer state that S. pedo is directly affected by fragmentation per se but rather emphasize how habitat reduction and connectivity loss can impact its conservation due to its limited mobility.
Comment 11: Line 58-60: Authors should add that S. pedo was considered a least concern species in Europe (Hochkirch et al. 2016), should provide the year of publication of the European Union Habitats Directive and the year of evaluation of the global conservation status by the IUCN.
Response 11: Clarified.
Comment 12 :Line 75: “in” is repeated twice. Delete one.
Response 12: corrected.
Comment 13:Line 76: Delete coma in “Alessandria, (Fig. 1)”.
Response 13: corrected.
Comment 14:Line 79-80: Add hornbeam-elm and chestnut trees latin names (as you did right before for Quercus pubescens).
Response 14: added.
Comment 15:Line 82: Replace “environments” by “habitats”.
Response 15: corrected.
Comment 16:Line 81-84: Please clarify “due to the cease of mowing, similar to abandoned agricultural and cultivated areas, which are mainly characterized by improved grasslands”. (I do not understand “similar to abandoned agricultural and cultivated areas” nor “improved grasslands”).
Response 16: We have revised the sentence for better clarity and have explicitly acknowledged that the term improved grasslands is derived from the cited literature.
Comment 17: Figure 1: Scale bars should be added and the figure legend should contain more details. Please indicate that the right part is a map of Italy with administrative province / region limits. Please add some localities to be used as geographic markers.
Response 17: Done.
Comment 18:Line 92: Please clearly indicate the number of S. pedo occurences obtained from Repetto et al. (2024) and the number of S. pedo occurences obtained from INaturalist.
Response 18: The number of S. pedo occurrences obtained from Repetto et al. (2024) and from iNaturalist are already indicated in the first phase of the Results section:
"From (Repetto et al., 2024), we considered nine sites in which S. pedo occurred and four in which our target species did not occur, while from the data available in iNaturalist, between April and July 2021-2024, we collected a total of 23 occurrences of S. pedo." now changed in "We considered a total of 34 S.pedo occurrence (nine from Repetto et al. [24] + 25 occurrences of S.pedo collected in our study area by eight observers and stored on the platform iNaturalist, www.inaturalist.org, accessed on 10 February 2025".
This information explicitly states the number of occurrences from both sources (9+25).
Comment 19: Line 98: I agree that the period between April and July is the period of maximal biological activity, however, I’m surprised that the authors did not search to obtain as many data as possible to feed their mathematical models/analysis and choose to discard the data pool obtained in August-September. Furthermore, if I were to expect dispersal within a species, I would expect it lately in the breeding season (Lakovic et al. 2024).
Response 19: We understood the concern and have carefully searched for additional occurrences of Saga p. in August and September, but unfortunately, our study area yielded no further findings in that years (see iNaturalist platform).
Comment 20: Beside the time period, shouldn’t you provide details about the geographic perimeter of your query on INaturalist ?
Response 20: We now specified in the STUDY SPECIES AND DATA paragraph that we collected S. pedo occurrences in an area of latitude ranging between 44.4° and 45° and longitude between 8.5° and 9.5°.
Comment 21: Line 101: Please clarify what you mean by “… to derive ‘observer-oriented’ (oo) pseudo-ab-101 sences [20–22] from iNaturalist (including both plants and animals)”.
Response 21: This sentence means that in our study we considered data from the iNaturalist platform, where observers of Saga p. record sightings of other species (including both plants and animals). "Observer-oriented" pseudo-absences refer to locations where observers have actively recorded other species but have not recorded our target species. This approach assumes that if the observers did not report the target species in these areas, it is likely absent there. Please, consider also the references cited in the text.
Comment 22: I guess that you assumed that your observers had always uploaded the presence of S. pedo in case they observed it. What if it they did not? Shouldn’t you provide more details about your use of non-Saga pedo observations? I think I understand that you considered as an absence of S. pedo a location where the observers reported non-Saga pedo observations but did not report S. pedo. However, I wonder if you shouldn’t have used a threshold value. For example, at a location where less than “x” (x=5? 10? …) non target species were reported, the search effort could be considered as too low to expect someone to find S. pedo even if the species is present.
Response 22: We understand the concerne of this Reviwer but we expect that the (few) observers of our target species would have recorded its occurrence in case of an observation and thus expect that where they collected other than the target species occurrence, they did not observe any S. pedo. However, we specified in the discussion section, as possible improvements of our approach, that researchers could additionally refine the list of pseudo-absences based on previously published papers on S. pedo co-occurring species.
Comment 23: Line 112: Could you please explain what is “Copernicus CLC+ Backbone 2018”? Aren’t you supposed to provide the DOI for this source? It is provided in the acknowledgments section but it would be more appropriate here.
Response 23: We specify in the text is actually a dataset "Land cover features were sourced from the Copernicus CLC+ Backbone 2018 dataset, also with a 10 m spatial resolution (https://land.copernicus.eu/en/products/clc-backbone , accessed on 15 August 2024),". The citation formats follow the guidelines for CLC+ citation.
Comment 24: Line 118: Add a coma after “In our study”.
Response 24: Corrected.
Comment 25: Line 125 and 132: Please choose how to spell “multicollinearity” / “multi-collinearity”
Response 25: Corrected.
Comment 26: Line 135: the whole chapter “data analysis” is very difficult to understand for a person who is not used to species distribution models.
Response 26: We did our best to simplify this and other methodological sections in this new version of our ms.
Comment 27: Line 141: Use “S. pedo” instead of Sp.
Response 27: Corrected.
Comment 28: Line 181-185: This should be included in the “Mat&Meth” section rather than in “Results” section.
Response 28: In others similar study this sentence is included in the Result section.
Comment 29: Line 183: You refer to figure 1 so the reader should expect to find the 23 S. pedo occurrences in your study site. However, actually there is no S. pedo occurrence shown in your figure 1.
Response 29: Corrected.
Comment 30: Line 184: As there are only seven observers, I think the authors should cite them. Currently they are not even thanked by name in the acknowledgements section.
Response 30: We understood the point and thus we thank thank ‘carlocabella-2020’, ’danirove’, ‘dgcurrywheel’, ‘emanuele_biggi’, ‘emanuelerepetto’, ‘germanoferrando’, ‘maracalvini’ and ‘maurizioappennino for uploading S. pedo occurrences on iNaturalist in the acknowledgements section.
Comment 31: Line 188: Is “reaming predictors” correct? Shouldn’t be “remaining predictors”?
Response 31: Corrected.
Comment 32: Line 196 and 198: Shouldn’t “the species occurrence” be replaced by “the S. pedo occurrence”?
Response 32: We modified here 'our target species occurrence' to avoid continuously repeat Saga pedo in all the lines.
Comment 33: Line 212, 213, 216: Replace “%”by “percentage”
Response 33: Corrected.
Comment 34: Line 217: I would not use “connectivity” but “habitat suitability”.
Response 34: We modified the capitation here.
Comment 35: Line 223: Shouldn’t “predicted species occurrence” be replaced by “predicted S. pedo occurrence”?
Response 35: Corrected.
Comment 36: I would have expected the “Landscape connectivity” to be the photographic negative of the “Probability of occurrence map” but it is not the case. Is it normal?
Response 36: We understand the point here and would like to clarify that we used the inverse (1 – probability of occurrence) as input file for Omniscape.jl and thus, the resulting landscape connectivity map is not a direct photographic negative of the "Probability of occurrence" map.
Comment 37: Line 235-236: This reference to the exploitation of natural resources is not appropriate. The authors should rather focus on the Mediterranean context evolution affecting open habitat and their affiliated species: the abandonment of itinerant pastoralism, the abandonment of agriculture in low productivity lands, first leading to an increase in habitats availability but now leading to its decrease due to afforestation.
Response 37: Clarified and corrected.
Comment 38: Line 242-243. I would rather insist on the precise definition of relative weight of the different habitat suitability parameters.
Comment 39: Line 255-256: S. pedo do not only have the potential to live in lowlands: it lives in lowlands when possible (see altitudinal distribution of the species in Occitanie-France : https://biodiv-occitanie.fr//espece/65680)
Response 39: We know that, but the our phrasing here is probably unclear. Clarified and corrected.
Comment 40: Line 278: I agree with authors that mathematical models and their “data-driven” conclusion are useful. However, I’m not so confident in models when they are “fed” with only 23 species observations.
Response 40: While we agree that having more locations would be better, we are confident that, as suggested by Erickson & Smith (2023), SDMs develop with 32 occurrences can provide reliable estimates of species distribution, especially, given the current status of Saga p. in our study area.
Erickson, K. D., & Smith, A. B. (2023). Modeling the rarest of the rare: A comparison between multi‐species distribution models, ensembles of small models, and single‐species models at extremely low sample sizes. Ecography, 2023(6), e06500.
Comment 41: References list.
Please check references list and ensure species latin names (Saga pedo, Popillia japonica) are in italic. Use capital letters only for the first letter of the genus name, not for the species name.
Response 41: Corrected.
Round 2
Reviewer 1 Report
Comments and Suggestions for Authors
I wish to thank authors for addressing my concerns. I am happy with the revised version of the manuscript.